# Genomic analysis of SARS-CoV-2 variants of concern circulating in Hawai'i to facilitate public-health policies

**David P. Maison**[1,2,3]*, **Sean B. Cleveland**[4,5], **Vivek R. Nerurkar**[1,2,3]*

**1** Department of Tropical Medicine, Medical Microbiology, and Pharmacology, University of Hawai'i—System, Honolulu, Hawai'i, United States of America, **2** Pacific Center for Emerging Infectious Diseases Research, University of Hawai'i—System, Honolulu, Hawai'i, United States of America, **3** John A. Burns School of Medicine, University of Hawai'i—System, Honolulu, Hawai'i, United States of America, **4** Hawai'i Data Science Institute, University of Hawai'i—System, Honolulu, Hawai'i, United States of America, **5** Information Technology Services—Cyberinfrastructure, University of Hawai'i—System, Honolulu, Hawai'i, United States of America

* davidm22@hawaii.edu (DPM); nerurkar@hawaii.edu (VRN)

**Data Availability Statement:** The viral genome sequences used in this publication are publicly available from GenBank (https://www.ncbi.nlm.nih.gov/sars-cov-2/) and GISAID (https://gisaid.org). Tables of acknowledgements for the genome

## Abstract

Using genomics, bioinformatics and statistics, herein we demonstrate the effect of statewide and nationwide quarantine on the introduction of SARS-CoV-2 variants of concern (VOC) in Hawai'i. To define the origins of introduced VOC, we analyzed 260 VOC sequences from Hawai'i, and 301,646 VOC sequences worldwide, deposited in the GenBank and global initiative on sharing all influenza data (GISAID), and constructed phylogenetic trees. The trees define the most recent common ancestor as the origin. Further, the multiple sequence alignment used to generate the phylogenetic trees identified the consensus single nucleotide polymorphisms in the VOC genomes. These consensus sequences allow for VOC comparison and identification of mutations of interest in relation to viral immune evasion and host immune activation. Of note is the P71L substitution within the E protein, the protein sensed by TLR2 to produce cytokines, found in the B.1.351 VOC may diminish the efficacy of some vaccines. Based on the phylogenetic trees, the B.1.1.7, B.1.351, B.1.427, and B.1.429 VOC have been introduced in Hawai'i multiple times since December 2020 from several definable geographic regions. From the first worldwide report of VOC in GenBank and GISAID, to the first arrival of VOC in Hawai'i, averages 320 days with quarantine, and 132 days without quarantine. As such, the effect of quarantine is shown to significantly affect the time to arrival of VOC in Hawai'i. Further, the collective 2020 quarantine of 43-states in the United States demonstrates a profound impact in delaying the arrival of VOC in states that did not practice quarantine, such as Utah. Our data demonstrates that at least 76% of all definable SARS-CoV-2 VOC have entered Hawai'i from California, with the B.1.351 variant in Hawai'i originating exclusively from the United Kingdom. These data provide a foundation for policymakers and public-health officials to apply precision public health genomics to real-world policies such as mandatory screening and quarantine.

sequences from GISAID are available at: https://github.com/dpmaison/Genomic-Analysis-of-SARS-CoV-2-Variants-of-Concern-Circulating-in-Hawai-i-to-Facilitate-Public-Healt.

**Funding:** This research was supported by a grant (P30GM114737-05) (V.R.N.) (https://hsrproject.nlm.nih.gov/view_hsrproj_record/20204464) from the Pacific Center for Emerging Infectious Diseases Research, COBRE and a grant (P20GM103466-20S1) (V.R.N) (https://taggs.hhs.gov/Detail/AwardDetail?arg_AwardNum=P20GM103466&arg_ProgOfficeCode=127) from the INBRE, National Institute of General Medical Sciences, NIH. Computation was supported by NSF grant #1920304 (S.B.C.) (https://www.nsf.gov/awardsearch/showAward?AWD_ID=1920304&HistoricalAwards=false) on the University of Hawai'i MANA High Performance Computing Cluster. The funders had no role in study design, data collection and analysis, decision to publish, or preparation of the manuscript.

**Competing interests:** The authors have declared that no competing interests exist.

# Introduction

Hawai'i has experienced unique epidemics within the coronavirus disease 2019 (COVID-19) pandemic, in that Pacific Islanders, which account for 4% of the population, once accounted for nearly 30% of COVID-19 cases [1]. Further, the Japanese population of Hawai'i currently accounts for 6% of the population and experiences 15% of COVID-19 cases. White persons, in contrast, account for 37% of the population and 25% of the cases [2]. As such, a heightened need exists to understand SARS-CoV-2 introduction into Hawai'i and the effect of public policy measures. Early in the pandemic, in an attempt to control the spread of severe acute respiratory syndrome coronavirus 2 (SARS-CoV-2), Hawai'i, like 42 other states in the United States, implemented a quarantine defined by "Stay-at-Home" orders. State-at-Home orders directed residents to stay inside homes except for essential needs and closed operations of non-essential businesses [3]. In addition to this public policy, more than 22,300 SARS-CoV-2 sequences submitted to GISAID and GenBank originate from Hawai'i to facilitate further studies.

The Pangolin/Phylogenetic Assignment of Named Global Outbreak (PANGO) Lineage nomenclature system developed by Rambaut and colleagues [4] for SARS-CoV-2 lineages has allowed for manageable and efficient partitioning of Hawai'i and worldwide sequences for the rapid determination of SARS-CoV-2 origin. The partitioning converts the vast number of sequences into smaller collections of pre-defined similar sequences. These can further generate multiple sequence alignments (MSA) to produce phylogenetic trees efficiently and at low cost.

Using the CDC-classified SARS-CoV-2 VOC (B.1.1.7, B.1.351, B.1.427, B.1.429, and P.1), identified in Hawai'i [5–7] as an example, we demonstrate a method to define the origin of SARS-CoV-2 lineages and VOC. This method works using either open-source or licensed software with either a personal computer or a supercomputer. Additionally, we evaluate the effect of quarantine in delaying the arrival of VOC. Using these methods and associated analysis to define the origin of introduction of VOC and determine the impact of quarantine, public-health officials can develop evidence-based policies to curtail the spread of VOC.

# Methods

## Defining the origin of SARS-CoV-2 VOC

Here, we present a method to define SARS-CoV-2 VOC origin. The architecture of this analysis is outlined in Fig 1 (DOI: dx.doi.org/10.17504/protocols.io.x54v9yqz4g3e/v1). We used 14 B.1.1.7, five B.1.351, 34 B.1.427, and 207 B.1.429 SARS-CoV-2 VOC sequences from Hawai'i deposited as of April 2, 2021, in the GenBank and global initiative on sharing all influenza data (GISAID) as an example. In brief, the method includes: 1) The lineage-defining sequences of SARS-CoV-2 Lineage A and Lineage B act as the most ancestral roots [4]. Lineage A (EPI_ISL_406801) is from GISAID, and Lineage B (MN908947) is from GenBank. 2) To identify lineages of interest in an area, filter GISAID by location (e.g.: North America/USA/Hawai'i) and download all sequences. For VOC with >10,000 sequences, GISAID sequences were downloaded in batches due to GISAID maximum download size. Similarly, all geographically similar sequences reported in GenBank were downloaded using the search term SARS-CoV-2 and state abbreviation (e.g., "SARS-CoV-2 HI") and the sequence length filter (20,000–40,000). 3) Combine the GISAID and GenBank sequences into one.fasta file using AliView, Geneious Prime 2021.0.3 (http://www.geneious.com), or a text editor, and assign lineages using Pangolin Lineage Assigner (pangolin.cog-uk.io) [4, 8–10]. 4) Download the results to Microsoft Excel, use advanced filter to copy unique records of lineages to a new column (ex: column M), then use COUNTIF (e.g., = COUNTIF($B$2:$B$1432,M2)) to determine

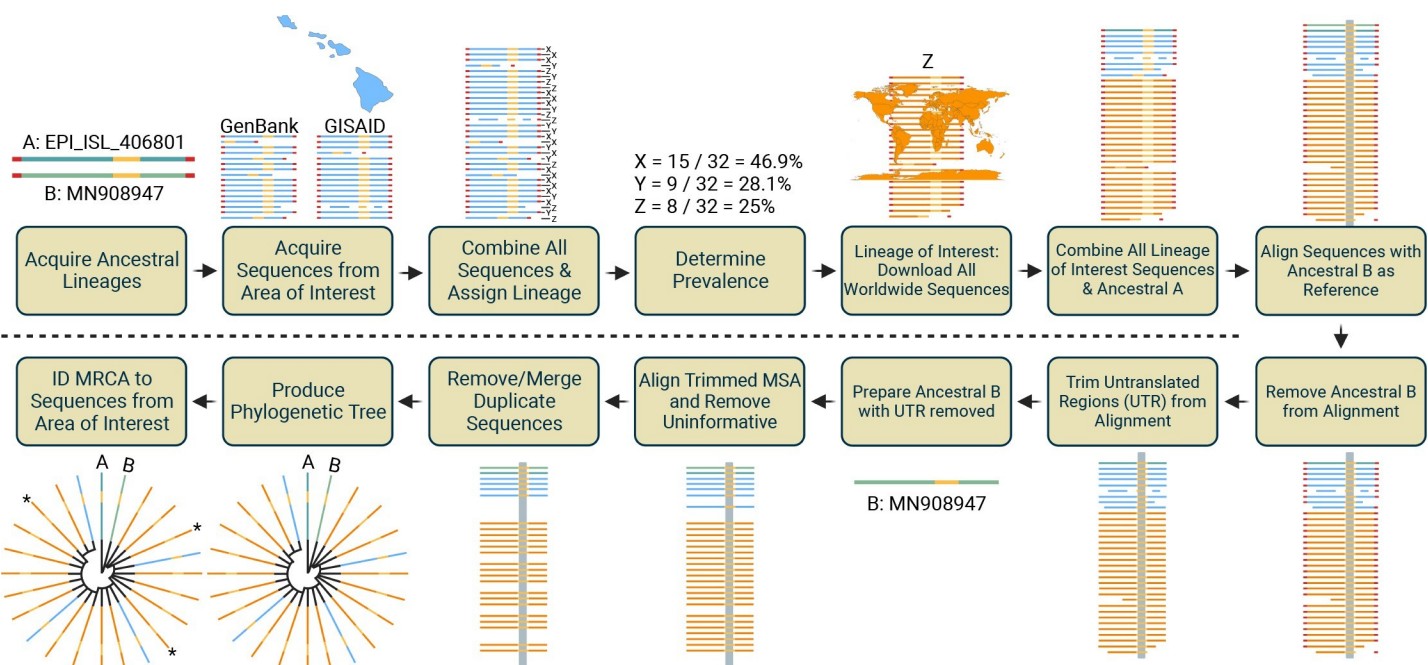

**Fig 1. Workflow to generate phylogenetic trees and to identify MRCA.** This figure outlines the 14 steps to define the origin of SARS-CoV-2 Variant of Concern (VOC) into an area of interest (demonstrated as Hawaii). Collectively, the workflow defines the VOC of a particular area, so research may identify a VOC of interest to conduct the workflow. Then all worldwide sequences of the selected VOC are aggregated into a Multiple Sequence Alignment (MSA). The MSA is modified to remove ambiguous and uninformative sequences and added to the ancestral lineages A (GISAID Accession # EPI_ISL_406801) and B (GenBank Accession # MN908947). Duplicate sequences are merged via their accession number (creating appendages), and the duplicates are removed from the MSA. An approximately maximum-likelihood phylogenetic tree is generated and rooted with ancestral lineage A. The identification (ID) of the most recent common ancestor (MRCA) of the sequences of interest defines the origin. Maps generated with the open-source "usmap" and"ggplot2" packages and R under GPL-3 and MIT+ license and are free to use (https://cran.r-project.org/web/packages/usmap/usmap.pdf)(https://cran.r-project.org/web/packages/ggplot2/index.html).

prevalence of each lineage. Alternatively, upload the results to Google Sheets and use the = UNIQUE command (e.g., = UNIQUE(B2:B1432) followed by the above COUNTIF command. 5) Identify sequences that are the lineage of interest (e.g.: B.1.429), from using Pangolin Lineage Assigner, and download those sequences individually from GenBank. Filter GISAID by the lineage of interest (e.g., B.1.429) and download all sequences. 6) Combine lineage of interest (B.1.429) GenBank sequences, GISAID (B.1.429) sequences, and EPI_ISL_406801 into one fasta file. 7) Align sequences using multiple alignment using fast Fourier transform (MAFFT) program or server (https://mafft.cbrc.jp/alignment/server/add_fragments.html?frommanualnov6) [11–13] with MN908947 as a reference and do not remove any uninformative sequences and all parameters set as "same as input." 8) Remove the newly added MN908947 sequence that MAFFT places at the beginning of the alignment using AliView, Geneious Prime, or a text editor. If not, the sRNA toolbox will remove the MN908947 sequence during the duplicate removal step, and Lineage B will not serve as an ancestral root in the phylogenetic tree. 9) Import MSA file into Geneious Prime or AliView [10], search for the orf1a 5' start of the entire alignment (5'-atggagagccttgtccctggtttca-3') and remove the 5' untranslated region (UTR) by deleting the upstream region (~265 bp) from the MSA. Next, search for ORF10 3' end (5'-tgtagttaactttaatctcacatag-3') and remove the entire 3' UTR by deleting the downstream region (~229 bp) from the MSA. 10) Create a duplicate file for the MN908947 sequence and remove the 5' UTR and 3' UTR from MN908947 as described above. 11) Using MAFFT, align the trimmed MSA with the trimmed MN908947 as a reference and delete sequences with uncalled nucleotides 'n'. Set the "remove

uninformative sequences" parameter in the MAFFT at >0%. 12) Using sRNAtoolbox program or server (https://arn.ugr.es/srnatoolbox/helper/removedup/) [14], load the updated alignment to remove duplicate sequences and merge identifications (also referred to as sequence accession numbers) of duplicates. This merger will create "appendages" in the phylogenetic tree where the sRNA toolbox will line up identical sequences together with equal signs (=). 13) Import the final alignment into Geneious Prime and create an approximately maximum-likelihood phylogenetic tree using the FastTree program [15]. Alternatively, FastTree can run as standalone software, and FastTreeMP is appropriate when multiple CPU cores/threads are available. 14) Root the tree with Lineage A (EPI_ISL_406801), which should then be the most recent common ancestor (MRCA) to Lineage B (MN908947) if performing phylogenetics on a Lineage B subgroup. Identify the MRCA of each sequence of interest.

For the B.1.1.7 VOC, which began with 272,732 sequences, we partitioned the sequences into seven sub-MSAs of ~50,000 sequences and performed the above method on each sub-MSA. After unambiguous sequences and duplicates were removed from each group, sub-MSA were recombined using AliView [10]. Duplicates were removed after each recombination of two sub-MSA, except for the final MSA due to size restrictions. Four sub-MSA were combined to create the final MSA. All sequences in the final MSA are unambiguous. However, there are likely duplicate sequences present. FastTreeMP generated the phylogenetic tree for B.1.1.7 in the University of Hawai'i MANA High-Performance Computing Cluster (HPC). All other steps, including the phylogenetic trees for B.1.351, B.1.427, and B.1.429, were done using a 2014 Apple MacBook Pro (2.6 GHz Intel Core i5, 8 GB RAM) or a Dell OptiPlex 3070 (3.0 GHz Intel Core i5, 8 GB RAM).

## Identifying the consensus of each VOC

To identify consensus SNPs of SARS-CoV-2 VOCs, assign Lineage B as the reference sequence, and use the Geneious Prime "Find Variations/SNPs" Annotate and Predict function to identify consensus SNPs. Input SNPs into the SnapGene (Insightful Science, snapgene.com) to identify the nucleotide and amino acid number and substitution as described previously [16].

## Evaluating the effect of quarantine

To test the hypothesis that the 67-day (2020-03-25 to 2020-05-31) [17] quarantine in Hawai'i, and the collective 43-state quarantine in the United States that occured from 2020-03-11 to 2020-06-16, significantly delayed the arrival of VOC, we partitioned the VOC into two categories. The first category of "quarantine" are those VOC (B.1.1.7, B.1.351, and B.1.429) that emerged worldwide before and during the 43-state collective mandatory quarantine was in effect. The second category of "post-quarantine" are those VOC (B.1.427 and P.1) that emerged worldwide after all quarantines were lifted. To determine the earliest collection date of each VOC worldwide, we analyzed all VOC (764,134) reported in the GISAID as of May 20, 2021. Since only 10 whole-genome sequences (WGS) of VOC B.1.351 have been reported from Hawai'i as of May 20, 2021, we analyzed the 10 first reported, and genetically distinct, of each VOC (B.1.429, B.1.427, B.1.1.7, B.1.351, and P.1) (total 47, since only 7 of the 10 B.1.351 were distinctive genetically) from Hawai'i as of May 20, 2021. For each VOC, we calculated the days between the first worldwide report and each of the first ten genetically distinct strains of VOC introduced in Hawai'i. We then compared the quarantine group to the post-quarantine group by days-to-arrival using an independent t-test in RStudio version 1.3.1093 (R version 4.0.3) and plotted with ggplot2 and ggstatsplot packages [18–20]. Further, we evaluated one of the seven states (AR, IA, ND, NE, SD, UT, WY) [21] that did not participate in the collective

43-state quarantine to determine if the delay in the arrival of VOC between quarantine and non-quarantine time periods exists for a state that did not quarantine. The nearest geographic state to Hawai'i of the remaining seven states which did not quarantine is Utah. Therefore, we evaluated Utah's first ten genetically distinct reports of each of the five VOC (Utah only had three strains of B.1.351 VOC, with two being unique (total 42)), as mentioned above, and compared them to the groups from Hawai'i.

## Results

As of April 02, 2021, 43 unique lineages were identified with Pangolin Lineage Assigner from the 1,431 total sequences deposited in the GenBank and GISAID from Hawai'i (Table 1). Based on this analysis, the B.1.429, B.1.427, B.1.1.7, and B.1.351 were 14.47%, 2.38%, 0.98%, and 0.35%, respectively, prevalent in Hawai'i. Moreover, the SARS-CoV-2 VOC, as classified by the CDC, were overall 18.2% (260/1,431) prevalent among all SARS-CoV-2 sequences in Hawai'i (Table 1). As of May 20, 2021, the VOC prevalence had increased overall to 45.2% (1,069/2,367) (Table 1). This increase includes the emergence of the P.1 VOC, a 11.78% increase in B.1.1.7, and a 14.13% increase in B.1.429 (Table 1).

### Variants of concern consensus

Fig 2 shows the phylogenetic analysis of all VOC prevalent in Hawai'i reported worldwide rooted with the Lineage A reference sequence (EPI_ISL_406801) [4]. These trees were generated using FastTree in MANA HPC and Geneious Prime [15]. Based on this analysis, 228 of the 260 (87.69%) VOC found in Hawai'i have identifiable origins. Fig 3 shows the states in the continental United States, as well as the countries worldwide, that were identified as being the source of the B.1.429, B.1.427, B.1.1.7, and B.1.351 SARS-CoV-2 VOC introductions into Hawai'i. The consensus (>90%) of the B.1.429 MSA, B.1.427 MSA, B.1.1.7 MSA, and B.1.351 MSA revealed 20, 16, 27, and 19, respectively, genomic mutations as compared to the MN908947 Lineage B reference sequence (Table 2).

### B.1.429 California VOC

In the GenBank, as of April 02, 2021, 21 of 97 Hawai'i sequences were of lineage B.1.429 as determined with the Pangolin Lineage Assigner. One-hundred eighty-six sequences in GISAID from Hawai'i were identified as the B.1.429 VOC. Total B.1.429 VOC reported worldwide in GISAID was 15,393 as determined by applying the GISAID lineage filter. Thus, the starting sequence count was 15,416 (21 GenBank + 15,393 GISAID + 2 lineage origin = 15,416). Of the 15,416 sequences, 11,648 sequences were removed for being uninformative and containing incomplete sequences. Further, the sRNAtoolbox server removed 944 sequences containing duplicate sequences and 15 sequences of duplicate ID. The final alignment of 2,809 (15,416–11,648 ambiguous—959 duplicate sequence and ID = 2,809) strains and subsequent phylogenetic analysis defined the origin of the B.1.429 variant introduced into Hawai'i (Fig 2A). Using this method, we were able to identify the origin of 183 of 207 B.1.429 sequences introduced into Hawai'i (Fig 3).

### B.1.427 California VOC

In the GenBank, as of April 02, 2021, 8 of 97 Hawai'i sequences were B.1.427 VOC as determined with the Pangolin Lineage Assigner. Twenty-six sequences in GISAID from Hawai'i were identified as the B.1.427 VOC. Total B.1.427 lineages reported worldwide in GISAID were 6,562 as determined by applying the GISAID lineage filter. Thus, the starting sequence

**Table 1. SARS-CoV-2 lineage prevalence in Hawai'i as of April 02, 2021 and May 20, 2021.**

| Lineage | VOC* Identifier | Prevalence of Lineages | | | VOC Reporting Dates | | |
|---|---|---|---|---|---|---|---|
| | | April 02, 2021 | May 20, 2021 | Δ% | First Reported VOC | Arrival of VOC in Hawaii | Δ (Days) |
| | | n (%) | n (%) | | | | |
| A.1 | - | 6 (0.42) | 6 (0.25) | -0.17 | - | - | - |
| A.2.2 | - | 1 (0.07) | 1 (0.04) | -0.03 | - | - | - |
| A.3 | - | 2 (0.14) | 2 (0.08) | -0.06 | - | - | - |
| B | - | 1 (0.07) | 1 (0.04) | -0.03 | - | - | - |
| B.1 | - | 66 (4.61) | 54 (2.28) | -2.33 | - | - | - |
| B.1.1 | - | 11 (0.77) | 11 (0.47) | -0.30 | - | - | - |
| B.1.1.207 | - | 1 (0.07) | 3 (0.13) | +0.06 | - | - | - |
| B.1.1.222 | - | 1 (0.07) | 1 (0.04) | -0.03 | - | - | - |
| B.1.1.304 | - | - | 2 (0.08) | +0.08 | - | - | - |
| B.1.1.316 | - | 1 (0.07) | 1 (0.04) | -0.03 | - | - | - |
| B.1.1.380 | - | 1 (0.07) | 1 (0.04) | -0.03 | - | - | - |
| B.1.1.416 | - | 7 (0.07) | 7 (0.3) | +0.23 | - | - | - |
| B.1.1.519 | - | 18 (1.26) | 43 (1.82) | +0.56 | - | - | - |
| **B.1.1.7** | **United Kingdom** | **14 (0.98)** | **302 (12.76)** | **+11.78** | **2020-02-07** | **2021-01-21** | **349** |
| B.1.108 | - | 2 (0.14) | 2 (0.08) | -0.06 | - | - | - |
| B.1.139 | - | 1 (0.07) | 1 (0.04) | -0.03 | - | - | - |
| B.1.160 | - | - | 9 (0.38) | +0.38 | - | - | - |
| B.1.2 | - | 171 (11.95) | 214 (9.04) | -2.91 | - | - | - |
| B.1.234 | - | 7 (0.49) | 7 (0.3) | -0.19 | - | - | - |
| B.1.241 | - | 2 (0.14) | 3 (0.13) | -0.01 | - | - | - |
| B.1.243 | - | 745 (52.06) | 751 (31.73) | -20.33 | - | - | - |
| B.1.265 | - | 2 (0.14) | 2 (0.08) | -0.06 | - | - | - |
| B.1.298 | - | 1 (0.07) | 1 (0.04) | -0.03 | - | - | - |
| B.1.340 | - | 1 (0.07) | 1 (0.04) | -0.03 | - | - | - |
| **B.1.351** | **South Africa** | **5 (0.35)** | **10 (0.42)** | **+0.07** | **2020-05-11** | **2021-02-16** | **281** |
| B.1.357 | - | 61 (4.26) | 61 (2.58) | -1.68 | - | - | - |
| B.1.36.8 | - | 5 (0.35) | 5 (0.21) | -0.14 | - | - | - |
| B.1.369 | - | 1 (0.07) | 1 (0.04) | -0.03 | - | - | - |
| B.1.37 | - | 1 (0.07) | 1 (0.04) | -0.03 | - | - | - |
| B.1.400 | - | 9 (0.63) | 9 (0.38) | -0.25 | - | - | - |
| B.1.413 | - | 4 (0.28) | 4 (0.17) | -0.11 | - | - | - |
| **B.1.427** | **California** | **34 (2.38)** | **46 (1.94)** | **-0.44** | **2020-09-17** | **2020-12-07** | **81** |
| **B.1.429** | **California** | **207 (14.47)** | **677 (28.6)** | **+14.13** | **2020-04-15** | **2020-12-31** | **260** |
| B.1.517 | - | 1 (0.07) | 1 (0.04) | -0.03 | - | - | - |
| B.1.526 | - | 3 (0.21) | 17 (0.72) | +0.51 | - | - | - |
| B.1.526.1 | - | - | 3 (0.13) | +0.13 | - | - | - |
| B.1.526.2 | - | - | 18 (0.76) | +0.76 | - | - | - |
| B.1.561 | - | 1 (0.07) | 8 (0.34) | +0.27 | - | - | - |
| B.1.568 | - | - | 3 (0.13) | +0.13 | - | - | - |
| B.1.575 | - | 1 (0.07) | 1 (0.04) | -0.03 | - | - | - |
| B.1.588 | - | 1 (0.07) | 1 (0.04) | -0.03 | - | - | - |
| B.1.595 | - | 4 (0.28) | 4 (0.17) | -0.11 | - | - | - |
| B.1.596 | - | 18 (1.26) | 24 (1.01) | -0.25 | - | - | - |
| B.1.601 | - | 1 (0.07) | 1 (0.04) | -0.03 | - | - | - |
| B.1.609 | - | 3 (0.21) | 3 (0.13) | -0.08 | - | - | - |

*(Continued)*

**Table 1.** (Continued)

| Lineage | VOC* Identifier | Prevalence of Lineages | | | VOC Reporting Dates | | |
|---|---|---|---|---|---|---|---|
| | | April 02, 2021 | May 20, 2021 | Δ% | First Reported VOC | Arrival of VOC in Hawaii | Δ (Days) |
| | | n (%) | n (%) | | | | |
| B.6 | - | 5 (0.35) | 5 (0.21) | -0.14 | - | - | - |
| **P.1** | **Brazil** | - | **34 (1.44)** | **+1.44** | **2020-11-03** | **2021-03-21** | **138** |
| P.2 | - | 2 (0.14) | 2 (0.08) | -0.06 | - | - | - |
| R.1 | - | 2 (0.14) | 2 (0.08) | -0.06 | - | - | - |
| Total | - | 1,431 | 2,367 | | | | |

*VOC = Variant of concern (depicted in bold)

count was 6,572 (8 GenBank + 6,562 GISAID + 2 lineage origin = 6,572). Of the 6,572 sequences, 5,273 sequences were removed for being ambiguous. Further, the sRNAtoolbox server removed 278 sequences containing duplicate sequences and 3 sequences of duplicate ID. One duplicate ID was created by the duplicate sequence merger. The final alignment of 1,019 strains (6,572–5,273 ambiguous—(281 duplicate sequence and ID—1 duplicate ID created by duplicate sequence merger) = 1,019) and subsequent phylogenetic analysis defined the origin of the B.1.427 variant introduced into Hawai'i (Fig 2B). Using this method, we were able to identify the origin of 22 of 34 B.1.427 sequences introduced into Hawai'i (Fig 3).

### B.1.1.7 United Kingdom VOC

In the GenBank, as of April 02, 2021, 0 of 97 Hawai'i sequences were of lineage B.1.1.7 as determined with the Pangolin Lineage Assigner. Fourteen strains in GISAID from Hawai'i were identified as the B.1.1.7 VOC. Total B.1.1.7 lineages reported worldwide in GISAID were 272,730 as determined by applying the GISAID lineage filter. Thus, the starting sequence count was 272,734 (272,730 GISAID sequences + 4 lineage origin). The aforementioned method was applied to each of the seven sub-MSA, thereby removing a total of 113,685 ambiguous sequences, and 75,590 duplicate sequences and ID. Twelve duplicate IDs were created by the duplicate sequence merger. The final alignment of 83,471 sequences (272,734–113,685 ambiguous—(75,590 duplicate sequence and ID—12 duplicate ID created by duplicate sequence merger) = 83,471) and subsequent phylogenetic analysis defined the origin of the B.1.1.7 VOC introduced into Hawai'i using phylogenetic analysis (Fig 2C). Using this method, we were able to identify the origin of 10 of 14 B.1.1.7 sequences introduced into Hawai'i (Fig 3).

### B.1.351 South Africa VOC

In the GenBank, as of April 02, 2021, 0 of 97 Hawai'i sequences were B.1.351 VOC as determined with the Pangolin Lineage Assigner. Five sequences in GISAID from Hawai'i were identified as the B.1.351 VOC. Total B.1.351 sequences reported worldwide in GISAID were 6,961 as determined by applying the GISAID lineage filter. Of the 6,963 sequences aligned, 5,758 sequences were removed for being uninformative and containing incomplete sequences. Further, the sRNAtoolbox server removed 373 sequences containing duplicate sequences and 55 sequences of duplicate ID. One duplicate ID was created by the duplicate sequence merger. The final alignment of 778 strains (6,961 GISAID + 2 lineage origin—5,758 ambiguous—(428 duplicate sequence and ID—1 duplicate ID created by duplicate sequence merger) = 778) and subsequent phylogenetic analysis defined the origin of the B.1.1.7 VOC introduced into

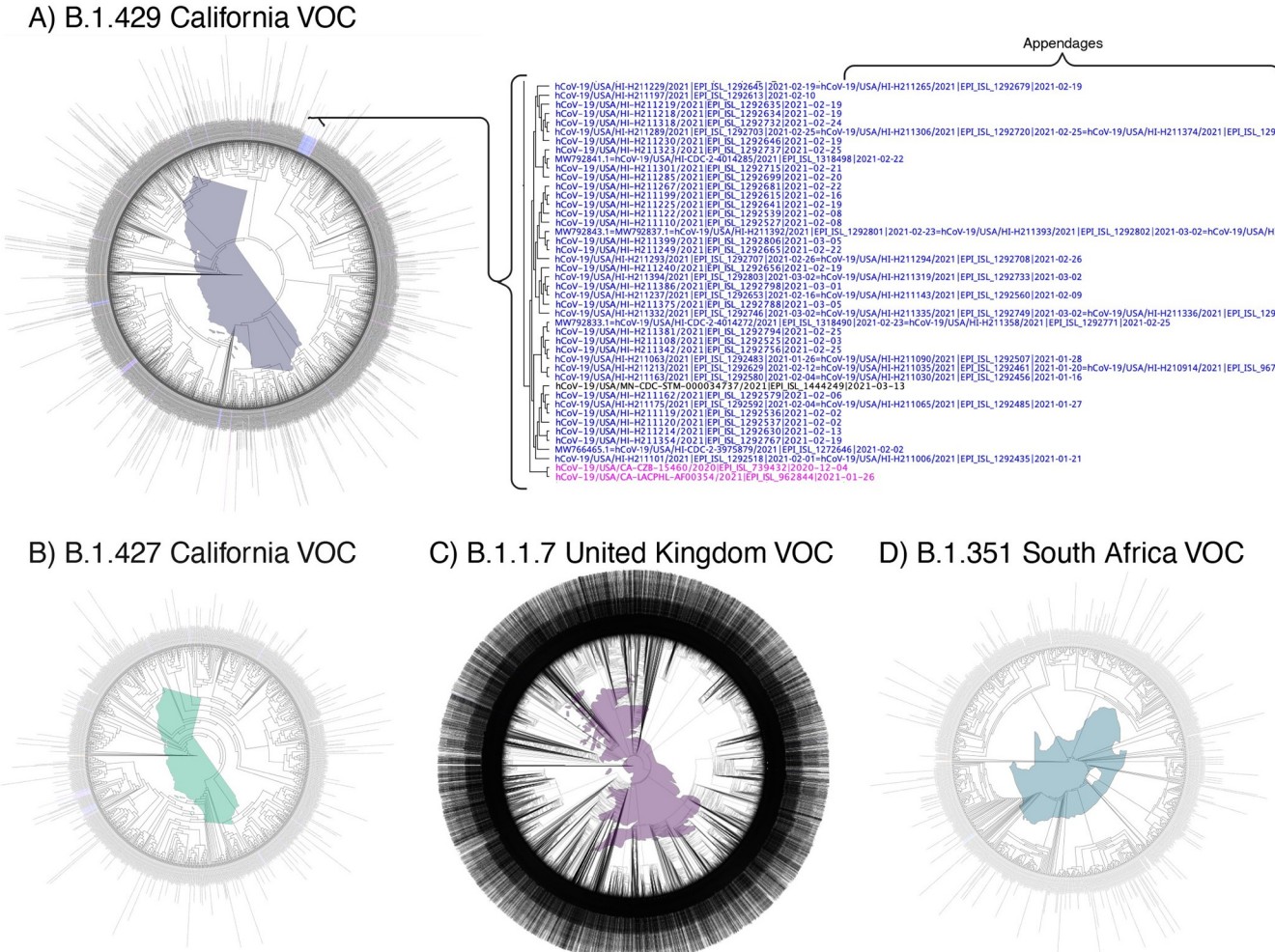

**Fig 2. Phylogenetic trees of worldwide SARS-CoV-2 variants of concerns.** The phylogenetic trees show the approximately maximum-likelihood trees generated by FastTree in Geneious Prime 2021.0.3 (http://www.geneious.com) and FastTreeMP in the University of Hawai'i MANA High Performance Computing Cluster. The trees were rooted to the SARS-CoV-2 Lineage A reference sequence (EPI_ISL_406801). Clusters and strains from Hawai'i are identified with the colored text (blue), and were evaluated for the most recent common ancestor to define the origin of Hawai'i strains (pink). Text shown in black indicates global variant strains not necessarily directly affiliated with variants found in Hawai'i. Appendage sequences designated by an equal sign (=) indicate identical sequences as generated by the sRNAtoolbox. A) Phylogenetic tree of B.1.429 California variant of concern (VOC) generated using Geneious Prime from 2,809 unambiguous and unique sequences. B) Phylogenetic tree of B.1.427 California VOC generated using Geneious Prime from 1,019 unique and unambiguous sequences. C) Phylogenetic tree of B.1.1.7 United Kingdom VOC generated with FastTreeMP in the MANA HPC using 83,471 unambiguous sequences. D) Phylogenetic tree of B.1.351 South Africa VOC generated in Geneious Prime using 778 unambiguous and unique sequences. Maps generated with the open-source "usmap" and"ggplot2" packages and R under GPL-3 and MIT+ license and are free to use (https://cran.r-project.org/web/packages/usmap/usmap.pdf)(https://cran.r-project.org/web/packages/ggplot2/index.html). Map editing was done in Adobe Photoshop 22.4.2, and the final figure was created with BioRender.com.

Hawai'i using phylogenetic analysis (Fig 2D). Using this method, we were able to identify the origin of all five B.1.351 sequences introduced into Hawai'i (Fig 3).

## High-quality sequencing

Several published sequences worldwide have missing nucleotides between the 5' UTR and 3' UTR, such as 41.7%, 82.7%, 80.2%, and 75.6% of B.1.1.7, B.1.351, B.1.427, and B.1.429 sequences, respectively. Therefore, these sequences are not useful for phylogenetic analysis.

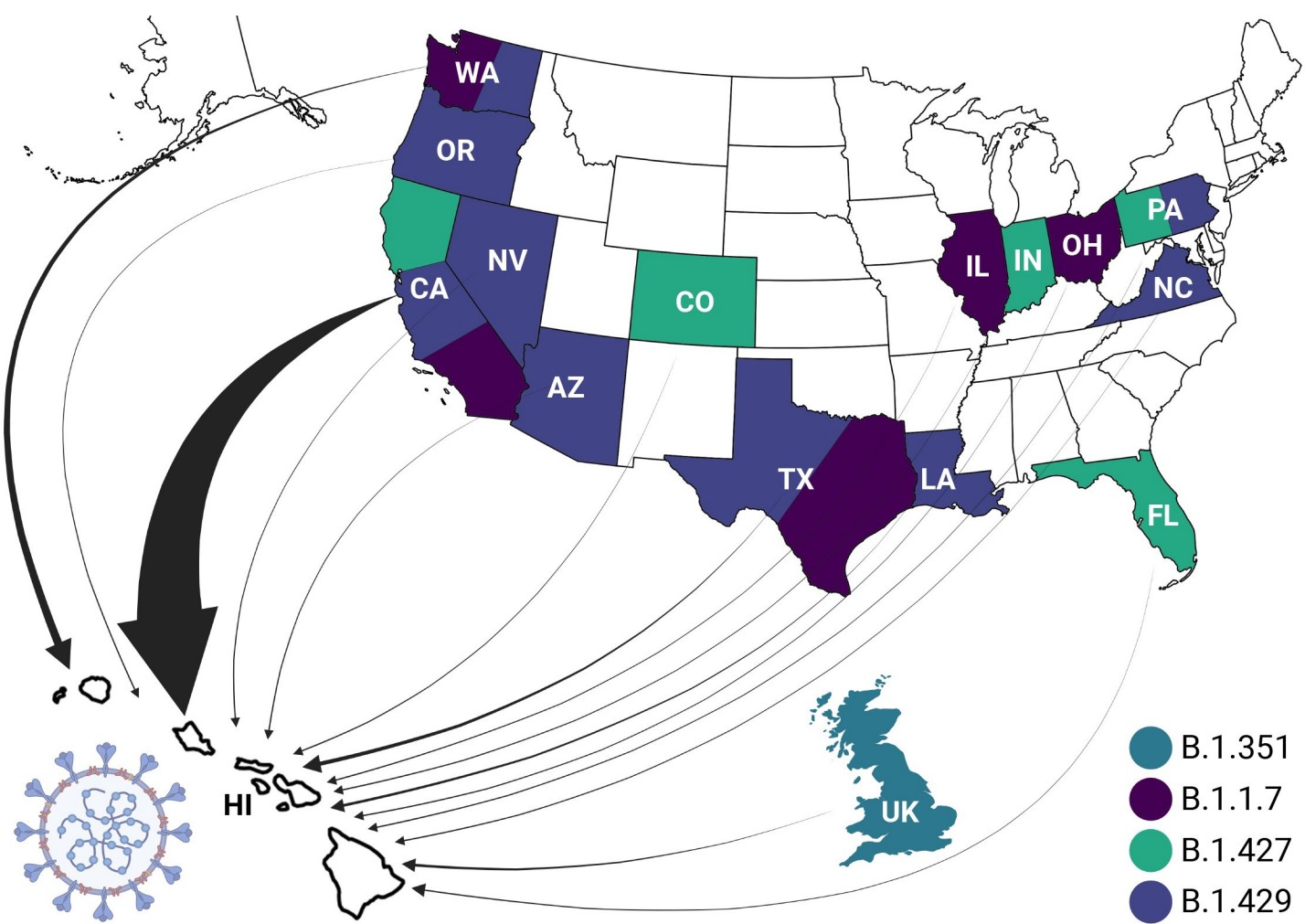

**Fig 3. Introduction of the SAR-CoV-2 variants of concern in Hawai'i.** This figure shows the states within the United States, and countries worldwide, from which the SARS-CoV-2 variants of concern (VOC) have been introduced into Hawai'i. Represented geographical locations are: California (CA), Washington (WA), Texas (TX), Louisiana (LA), the United Kingdom (UK), Pennsylvania (PA), Colorado (CO), Florida (FL), Arizona (AZ), Illinois (IL), Indiana (IN), Nevada (NV), North Carolina (NC), Ohio (OH) and Oregon (OR). Blue represents the origin of B.1.351 VOC, yellow represents the origin of B.1.1.7 VOC, pink represents the origin of B.1.427 VOC, and the green represents the origin of B.1.429 VOC. The width of the arrows is proportional to the abundance of introductions of VOC from various geographic areas. Maps generated with the open-source "usmap" and "ggplot2" packages and R under GPL-3 and MIT+ license and are free to use (https://cran.r-project.org/web/packages/usmap/usmap.pdf)(https://cran.r-project.org/web/packages/ggplot2/index.html). Map editing was done in Adobe Photoshop 22.4.2 and the final figure was created with Biorender.com.

**Quarantine.** As of May 20, 2021, the first reported collection dates worldwide for B.1.429, B.1.427, B.1.1.7, B.1.351, and P.1 VOC, respectively, are 2020-04-15 [22], 2020-09-17 [23], 2020-02-07 [24], 2020-05-11 [25], and 2020-11-03 [26]. In Hawai'i, the first reported collection dates for B.1.429, B.1.427, B.1.1.7, B.1.351, and P.1 VOC were on 2020-12-31, 2020-12-07, 2021-01-21, 2021-02-16, and 2021-03-21, respectively [27]. From first reported worldwide collection date to first reported collection date in Hawai'i, for the B.1.429, B.1.427, B.1.1.7, B.1.351, and P.1 VOC, are 260, 81, 349, 281, and 138 days, respectively (Fig 4).

Partitioning days to VOC arrival in Hawai'i into quarantine (2020-03-11 to 2020-06-16) ($M$ = 320 days, SD = 45) and post-quarantine (2020-06-16 to 2021-05-20) ($M$ = 132 days, SD = 21) time periods demonstrates that quarantine significantly delayed the arrival of VOC to Hawai'i, $t(45) = 17.38$, $p = 1.38e-21$ (Fig 5A). Utah, a non-quarantine state, also demonstrated

**Table 2. Single nucleotide polymorphisms, amino acid substitutions, and deletions among SARS-CoV-2 variants of concern circulating in Hawai'i.**

| Gene or region | Nucleotides and Amino Acids | | | | VOC Circulating in Hawai'i | | | |
|---|---|---|---|---|---|---|---|---|
| | Nucleotide Loci | Nucleotide Change | Amino Acid Position | Amino Acid Change | B.1.427 | B.1.429 | B.1.1.7 | B.1.351 |
| **orf1ab** | 913 | C → T | 216 | - | | | ■ | |
| | 1,059 | C → T | 265 | Thr → Ile | ■ | | | ■ |
| | 2,395 | C → T | 710 | - | | ■ | | |
| | 2,597 | T → C | 778 | - | | ■ | | |
| | 3,037 | C → T | 924 | - | ■ | ■ | ■ | ■ |
| | 3,267 | C → T | 1,001 | Thr → Ile | | | ■ | |
| | 5,230 | G → T | 1,655 | Lys → Asn | | | | ■ |
| | 5,388 | C → A | 1,708 | Ala → Asp | | | ■ | |
| | 5,986 | C → T | 1,907 | - | | | ■ | |
| | 6,954 | T → C | 2,230 | Ile → Thr | | | ■ | |
| | 8,947 | C → T | 2,894 | - | | ■ | | |
| | 9,738 | G → C | 3,158 | Ser → Thr | ■ | | | |
| | 10,323 | A → G | 3,353 | Lys → Arg | | | | ■ |
| | 11,288–11,296 | Δ | 3,675–3,677 | ΔSerΔGlyΔPhe | | | ■ | ■ |
| | 12,100 | C → T | 3,945 | - | | ■ | | |
| | 12,878 | A → G | 4,205 | Ile → Val | | ■ | | |
| | 13,713 | G → A | 4,483 | - | ■ | | | |
| | 14,408 | C → T | 4,715 | Pro → Leu | ■ | ■ | ■ | ■ |
| | 14,676 | C → T | 4,804 | - | | | ■ | |
| | 15,279 | C → T | 5,005 | - | | | ■ | |
| | 16,176 | T → C | 5,304 | - | | | ■ | |
| | 16,394 | C → T | 5,377 | Pro → Leu | ■ | | | |
| | 17,014 | G → T | 5,584 | Asp → Tyr | ■ | ■ | | |
| **S** | 21,600 | G → T | 13 | Ser → Ile | ■ | ■ | | |
| | 21,765–21,770 | Δ | 69–70 | ΔHisΔVal | | | ■ | |
| | 21,801 | A → C | 80 | Asp → Ala | | | | ■ |
| | 21,991–21,993 | Δ | 144 | ΔTyr | | | ■ | |
| | 22,018 | G → T | 152 | Trp → Cys | ■ | ■ | | |
| | 22,206 | A → G | 215 | Asp → Gly | | | | ■ |
| | 22,281–22,289 | Δ | 242–244 | ΔLeuΔAlaΔLeu | | | | ■ |
| | 22,813 | G → T | 417 | Lys → Asn | | | | ■ |
| | 22,917 | T → G | 452 | Leu → Arg | ■ | ■ | | |
| | 23,012 | G → A | 484 | Glu → Lys | | | | ■ |
| | 23,063 | A → T | 501 | Asn → Tyr | | | ■ | ■ |
| | 23,271 | C → A | 570 | Ala → Asp | | | ■ | |
| | 23,403 | A → G | 614 | Asp → Gly | ■ | ■ | ■ | ■ |
| | 23,604 | C → A | 681 | Pro → His | | | ■ | |
| | 23,664 | C → T | 701 | Ala → Val | | | | ■ |
| | 23,709 | C → T | 716 | Thr → Ile | | | ■ | |
| | 24,349 | T → C | 929 | - | | ■ | | |
| | 24,506 | T → G | 982 | Ser → Ala | | | ■ | |
| | 24,914 | G → C | 1,118 | Asp → His | | | ■ | |
| **ORF3a** | 25,563 | G → T | 57 | Gln → His | ■ | ■ | | ■ |
| | 25,904 | C → T | 171 | Ser → Leu | | | | ■ |
| **E** | 26,456 | C → T | 71 | Pro → Leu | | | | ■ |
| **M** | 26,681 | C → T | 53 | - | ■ | ■ | | |

*(Continued)*

**Table 2.** (Continued)

| Gene or region | Nucleotide Loci | Nucleotide Change | Amino Acid Position | Amino Acid Change | B.1.427 | B.1.429 | B.1.1.7 | B.1.351 |
|---|---|---|---|---|---|---|---|---|
| | | | | **Nucleotides and Amino Acids** | | **VOC Circulating in Hawai'i** | | |
| **ORF7b/ORF8 intron** | 27,890 | G → T | - | - | | ■ | | |
| **ORF8** | 27,972 | C → T | 27 | Gln → stop | | | ■ | |
| | 28,048 | G → T | 52 | Arg → Ile | | | ■ | |
| | 28,111 | A → G | 73 | Tyr → Cys | | | ■ | |
| | 28,253 | C → T | 120 | - | | | | ■ |
| **ORF8/N intron** | 28,271 | Δ | - | - | | | ■ | |
| | 28,272 | A → T | - | - | ■ | ■ | | |
| **N** | 28,280–28,282 | GAT → CTA | 3 | Asp → Leu | | | ■ | |
| | 28,881–28,883 | GGG → AAC | 203–204 | ArgGly → LysArg | | | ■ | |
| | 28,887 | C → T | 205 | Thr → Ile | ■ | | | ■ |
| | 28,977 | C → T | 235 | Ser → Phe | | | ■ | |
| | 29,362 | C → T | 363 | - | ■ | ■ | | |

Consensus determined for B.1.427 VOC, B.1.429 VOC, B.1.1.7 VOC, and B.1.351 VOC from multiple sequence alignment of 1,019, 2,809, 83,471, and 778 strains, respectively. Mutations present in the corresponding VOC are shaded with black color.

difference in days to VOC arrival between the time period defined by the collective quarantine of the 43 states (2020-03-11 to 2020-06-16) ($M$ = 285 days, SD = 71) when compared to the nationwide no-quarantine time period (2020-06-16 to 2021-05-20) ($M$ = 116 days, SD = 40), t(40) = 9.37, $p$ = 1.2e-11 (Fig 5B). Comparing the group defined by the collective 43-state quarantine demonstrates that VOC arrival in Hawai'i ($M$ = 320 days, SD = 45) was delayed when compared to Utah ($M$ = 285 days, SD = 71), t(47) = 2.11, $p$ = 0.04 (Fig 5C). Comparing the group defined by the time period following the collective 43-state quarantine demonstrates no significant difference in days to VOC arrival between Hawai'i ($M$ = 132 days, SD = 21) when compared to Utah ($M$ = 116 days, SD = 40), t(38) = 1.59, $p$ = 0.12 (Fig 5D).

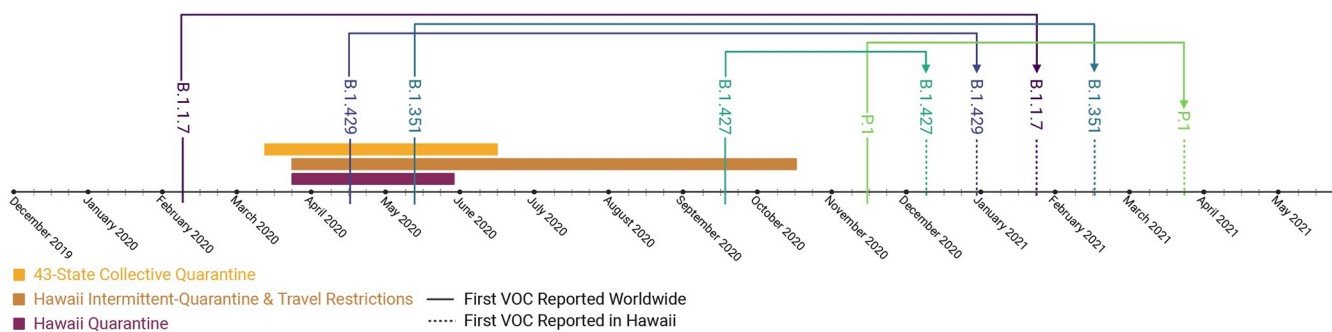

**Fig 4. Emergence of SARS-CoV-2 VOC timeline.** This figure shows the timeline of the emergence of SARS-CoV-2 variants of concern (VOC) with respect to Hawai'i. The x-axis is a calendar from December 2019 through May 20, 2021. Solid lines with the corresponding VOC name (B.1.1.7, B.1.429, B.1.351, B.1.427, and P.1) depict the worldwide emergence dates of the VOC. Dashed lines with the corresponding VOC name depict the first reported VOC date in Hawai'i. Dates of quarantine in Hawai'i (2020-03-22–2020-05-31), dates of intermittent quarantine in Hawai'i and trans-Pacific travel restrictions (2020-03-22–2020-10-15), and dates of the collective 43-state quarantine (2020-03-11 to 2020-06-16) are shown in colored boxes. As of May 20, 2021, the first reported collection dates worldwide for B.1.429, B.1.427, B.1.1.7, B.1.351, and P.1 VOC, respectively, are 2020-04-15, 2020-09-17, 2020-02-07, 2020-05-11, and 2020-11-03. In Hawai'i, the first reported collection dates for B.1.429, B.1.427, B.1.1.7, B.1.351, and P.1 VOC were on 2020-12-31, 2020-12-07, 2021-01-21, 2021-02-16, and 2021-03-21, respectively. The figure was created with Biorender.com.

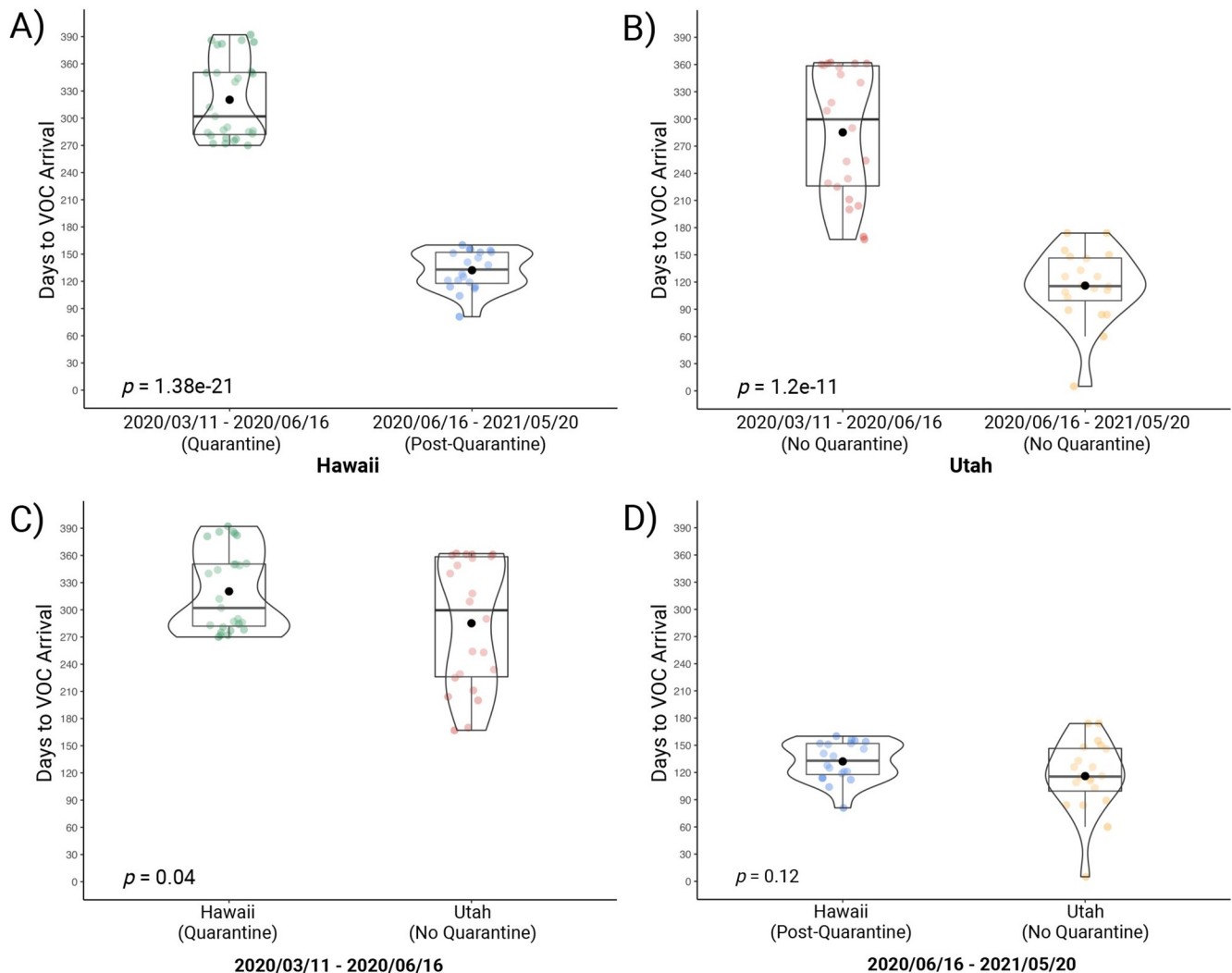

**Fig 5. Post-quarantine days to VOC arrival vs. quarantine days to VOC arrival in Hawai'i and Utah as of May, 20, 2021.** This figure shows the days-to-arrival of variants of concern (VOC) compared to VOC categorized as 'quarantine' and 'post-quarantine.' Depicted as a box-and-whisker/violin plot displaying median (line), mean (black circle), interquartile range, minimum, and maximum. Quarantine VOC are those VOC that emerged worldwide during the 43-state collective quarantine in the United States (2020-03-11 to 2020-06-16). Post-quarantine VOC are those VOC that emerged worldwide following the collective quarantine in the United States (2020-06-16 to 2021-05-20). The comparison was done using an independent t-test. A) Partitioning days to VOC arrival in Hawai'i into quarantine ($M$ = 320 days, SD = 45) and post-quarantine ($M$ = 132 days, SD = 21) time periods demonstrates that quarantine delayed the arrival of VOC to Hawai'i (t(45) = 17.38, $p$ = 1.38e-21). B) For Utah, a state that did not quarantine during the 43-state collective quarantine, a difference is also demonstrated between the two time periods (quarantine $M$ = 285 days, SD = 71; post-quarantine $M$ = 116 days, SD = 40) regarding time to VOC arrival (t(40) = 9.37, $p$ = 1.2e-11). C) Comparing Hawai'i and Utah by the quarantine time period demonstrates that Hawai'i ($M$ = 320 days, SD = 45), a quarantine state, saw a greater delay in days to VOC arrival compared to Utah ($M$ = 285 days, SD = 71), a non-quarantine state (t(47) = 2.11, $p$ = 0.04). D) Comparing Hawai'i and Utah by the post-quarantine time period demonstrates no difference between Hawai'i ($M$ = 132 days, SD = 40) and Utah ($M$ = 116 days, SD = 40) (t(38) = 1.59, $p$ = 0.12) in days to VOC arrival. Created with RStudio version 1.3.1093 (R version 4.0.3) using ggplot2 and ggstatsplot packages. The final figure was created with BioRender.com.

## Discussion

Precision public health genomics has been a tool in past outbreaks that has yet to be applied for the COVID-19 pandemic. These data and the method serve as a foundation for policy-makers to apply precision public health genomics tools by discerning trends related to the source of SARS-CoV-2 introductions. By identifying the origin of SARS-CoV-2, policies can be reasonably constructed with evidence-based decisions.

## The origins of variants of concern in Hawai'i

Fourteen geographical locations are definable as the origin of SARS-CoV-2 VOC in Hawai'i. From the most to the least VOC introductions in Hawai'i, these locations are California (174 introduced VOC), Washington (19), Texas (10), Louisiana (6), United Kingdom (5), Pennsylvania (4), Colorado (2), Florida (1), Arizona (1), Illinois (1), Indiana (1), Nevada (1), North Carolina (1), Ohio (1) and Oregon (1).

The MRCA branch to the sequences from Hawai'i indicates that the B.1.429, B.1.427, and B.1.1.7 VOCs were introduced into Hawai'i independently at different times from the aforementioned states in the continental United States. The B.1.351 VOC was introduced to Hawai'i from the United Kingdom. The overwhelming majority of the VOC entering Hawai'i are exclusively from California. In 2020, 27% of all travelers to Hawai'i originated from California, with 53% coming from the West Coast. Further, Hawai'i residents traveled to the West Coast, specifically Las Vegas, Nevada [1, 28, 29].

## Defining VOC using genetic characterization

As an effect of performing the origin defining method, described in the methods section, the MSA of all unambiguous VOC genomes can characterize the genome of VOC. Additionally, with this method, we can robustly compare the genomic similarities and differences between VOC. Within the S gene of the B.1.1.7 VOC, there are nine nucleotide and amino acid changes compared to the wild-type. These are Δ69–70, ΔY144, N501Y, A570D, D614G, P681H, T716I, S982A, and D1118H. Within the S gene of the B.1.427 and B.1.429 VOC, there is a consensus of four non-synonymous amino acid substitutions: S13I, W152C, L452R, D614G. Also, the S gene of the B.1.351 VOC encodes eight substitutions or deletions: D80A, D215G, Δ242–244, K417N, E484K, N501Y, D614G, and A701V. Further, the B.1.351 contains a substitution in the E gene (P71L) that results in the slightly stabilizing, based on the changes of Gibbs free energy, loss of a proline in the envelope protein [30]. The envelope protein was recently shown to interact with TLR2 and initiate inflammatory response [31]. Prolines are known to be involved in beta-turns, and the P71L substitution could significantly change the secondary and tertiary protein structures. This proline loss is striking for vaccines, since some vaccines effective against wild-type SARS-CoV-2, have diminished efficacy against the B.1.351 variant [32, 33].

Efforts to understand variants still focus primarily on identifying the effect of individual mutations and substitutions. Many of the other mutations have yet to be evaluated experimentally, either individually or in concert with the D614G substitution. The ubiquitous D614G substitution increases the fitness of SARS-CoV-2, even at the cost of increased susceptibility to neutralization [34–36]. As such, SARS-CoV-2 has been evolving, and substitutions in the B.1.351 VOC and P.2 variant of interest (VOI), such as the E484K, are shown to confer resistance to neutralization and exhibit 50% increased transmission [37, 38]. Recent publications focused on L452R mutation demonstrate significant decrease in the effectiveness of mAb treatments and neutralization by convalescent and vaccine sera [39, 40]. Other mutated genes within VOC are orf1ab, ORF3a, M, ORF8, and N, as well as two introns. Mutations and substitutions in these genes and proteins, annotated in Table 2, are enigmatic, and warrant further studies. Conclusively, what is certain is that tracking the spread of these VOC, and determining the effects of their substitutions, is paramount in the effort to control the pandemic.

## Ambiguous sequences

This method demonstrates the need for high-quality sequencing and the need for enrichment and deep coverage. For example, the first B.1.429 sequence deposited from Hawai'i was from a sample collected on December 31, 2020 (EPI_ISL_967766) is presently unusable. Without

resequencing the whole genome or filling in with Sanger sequencing, this sequence is currently not of use in phylogenetics and origin determination due to ambiguous nucleotides in the S gene. While uninformative sequences may be useful for tracking the emergence of individual mutations, they are not useful in tracking VOC.

## Public policy recommendations and impacts

Precision public health genomics is a public health policy tool to track the spread of viruses. In the age of fast-evolving digital information, precision public health genomics became prominent during the West Africa Ebola outbreak from 2014–2016 [41]. This tool has not been efficiently and effectively used during the COVID-19 pandemic due to the overwhelming worldwide sequencing effort. A testament to this effort is the deposition of over 18 million SARS-CoV-2 WGS in the GenBank and GISAID since January 2020. For precision public health genomics to be effective during the COVID-19 pandemic, high-throughput sequencing and high-speed, low-cost sequence data analysis, and robust phylogenetics are necessary. Also, a fast, effective, consistent, and economical method is required to analyze the vast amount of SARS-CoV-2 sequences and determine the origin of SARS-CoV-2 VOC and lineages in populations worldwide.

As the scientific community continues to understand the vaccine and healthcare consequences of VOC, of crucial importance is to control and limit the spread of these VOC. Policy-makers should first ascertain the source of the spread before they can control and limit the spread of future VOC. By understanding the source responsible for the highest number of cases, policy-makers can look at interactions between that area and the host area, identify the reasons for the spread, and address those with appropriate measures both in the present and future COVID-19 waves. After policy-makers contain the source, healthcare providers will treat patients with the most effective therapy, and vaccines will uphold their efficacy. However, without such precision public health genomics in practice, society risks losing progress in the fight against this pandemic. As only 5.4% of the global population is fully vaccinated against SARS-CoV-2 as of May 28, 2021 [42], the possibility of new infections, even in vaccinated populations, is ever present without appropriate measures to deter the spread and evolution of SARS-CoV-2. To exemplify this healthcare point, the B.1.427 and B.1.429 are resistant to specific monoclonal antibody (mAb) therapies [40, 43]. Clinically, of treatment importance, the Food and Drug Administration and California Department of Health and Human Services have stopped using Bamlanivimab due to reduced clinical activity against the B.1.427 and B.1.429 variants [43]. Furthermore, the B.1.351 VOC demonstrates some resistance to mAbs, convalescent sera, and vaccine sera neutralizing antibodies, and the B.1.1.7 is resistant to some mAbs [44].

Much of Hawai'i was under a mandatory stay-at-home order (2020-03-22–2020-05-31) from the Governor of Hawai'i's Third [45] (2020-03-22) through Eighth [46] (2020-05-18) COVID-19 Proclamation's, with intermittent and inter-island quarantines continuing through the Thirteenth [47] (2020-09-23) Proclamation. Trans-Pacific travelers were no longer required to quarantine as of October 15, 2020 [48]. Of interest is that even though VOC were circulating worldwide, none entered Hawai'i during the stay-at-home order or intermittent quarantines; all entered after the conclusion of the Thirteenth Proclamation and the reinstatement of Trans-Pacific travel. Furthermore, the average time it took for the VOC to arrive into Hawai'i decreased from an average of 320 days to 132 days when compared between the quarantine and post-quarantine groups, respectively. The data here argues in favor of the success of the quarantine here in Hawai'i in preventing the influx of the SARS-CoV-2 VOC. Further, from 2020-03-11 to 2020-06-16, 43 states participated in a collective nationwide quarantine

averaging 46 days per state, although there was no point in time where all 43 states were under simultaneous quarantine. We observed that even the seven states not participating in quarantine were able to significantly delay arrival of VOC, analogous to the "concept of herd immunity." Furthermore, in the post-collective-quarantine period (2020/06/16–2021/05/20), wherein no states were practicing quarantine, there was no significant difference in days to VOC arrival. This finding supports the data that quarantine (2020/03/11–2020/06/16) indeed delayed the arrival of VOC. While individual states benefit from separate quarantine, the data here indicate the success and impact of collective quarantine of the 43-states within the United States.

As the purpose of this article is to facilitate evidence based public-health policy, the vast number of VOC (194/228) entering Hawai'i from the West Coast of the United States is alarming and warrants policies directed at controlling the spread of the VOC in the Hawaiian Islands.

From the analysis of the SARS-CoV-2 sequence data, a policy-maker could reasonably consider focusing on additional screening, contact tracing, and quarantine efforts among visitors and residents arriving from and traveling to the West Coast of the continental United States. In terms of public policies and precision public health genomics, this information can direct funding into scientific studies evaluating the effects of mutations prevalent in specific populations, particularly mutations within genes that affect SARS-CoV-2 immunogenic epitopes. Policies should encourage research focusing on developing pseudoviruses [49]. and infectious clones [50] to evaluate kinetics, virulence, anatomical localization, transmission, and neutralization by mAbs, convalescent sera, and vaccine sera. Funding to conduct the aforementioned research should be directed at local and national levels.

## Limitations

As SARS-CoV-2 sequences continue to be submitted retrospectively, these data will evolve. As a tool for precision public health genomics, the highest value is the trends that this method elucidates. The quarantine data will also change, this analysis is a snapshot in time.

## Conclusions

These methods demonstrate the ability of precision public health genomics to identify the origin of SARS-CoV-2, the success of quarantine in Hawai'i, and the concern of emerging VOC. The conclusion from defining the origin of VOC in Hawai'i is that California is the primary source of VOC circulating in Hawai'i. Additional screening and quarantining of the travelers from California while vacationing in Hawai'i will protect the local population from evasive SARS-CoV-2 VOC. A tool was needed to evaluate and make use of the vast worldwide sequencing effort and the tool herein fills that need. Moreover, our methodology demonstrates the ability of sequencing and phylogenetic analysis to provide precision public health genomics in policy-making decisions. As SARS-CoV-2 VOC spreads asymptomatically across the United States and worldwide, it is essential to use fast and accurate SARS-CoV-2 VOC, lineage, and origin assignment for making evidence-based public-policy decisions.

## Acknowledgments

We thank Dr. Vedbar Khadka for assistance with identifying appropriate statistical tests to be used in this study. The viral genome sequences used in this publication are publicly available from GenBank (https://www.ncbi.nlm.nih.gov/sars-cov-2/) and GISAID (https://gisaid.org). Tables of acknowledgments for the genome sequences from GISAID are available at: https://

github.com/dpmaison/Genomic-Analysis-of-SARS-CoV-2-Variants-of-Concern-Circulating-in-Hawai-i-to-Facilitate-Public-Healt.

## Author Contributions

**Conceptualization:** David P. Maison, Vivek R. Nerurkar.

**Data curation:** David P. Maison.

**Formal analysis:** David P. Maison, Sean B. Cleveland, Vivek R. Nerurkar.

**Funding acquisition:** Vivek R. Nerurkar.

**Investigation:** Vivek R. Nerurkar.

**Methodology:** David P. Maison, Vivek R. Nerurkar.

**Project administration:** Vivek R. Nerurkar.

**Resources:** David P. Maison, Vivek R. Nerurkar.

**Software:** David P. Maison, Sean B. Cleveland.

**Validation:** David P. Maison, Vivek R. Nerurkar.

**Visualization:** David P. Maison.

**Writing – original draft:** David P. Maison, Vivek R. Nerurkar.

**Writing – review & editing:** David P. Maison, Sean B. Cleveland, Vivek R. Nerurkar.

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
