## [Decision Letter · Decision Letter 0]

15 Jul 2022

PONE-D-21-19856

Genomic Analysis of SARS-CoV-2 Variants of Concern Circulating in Hawai’i to Facilitate Public-Health Policies

PLOS ONE

Dear Dr. Maison,

Thank you for submitting your manuscript to PLOS ONE. After careful consideration, we feel that it has merit but does not fully meet PLOS ONE’s publication criteria as it currently stands. Therefore, we invite you to submit a revised version of the manuscript that addresses the points raised during the review process.

We look forward to receiving your revised manuscript.

Kind regards,

Ming Zhang

Academic Editor

PLOS ONE

Journal Requirements:

2.  We note that Figure 2 in your submission contain map images which may be copyrighted. All PLOS content is published under the Creative Commons Attribution License (CC BY 4.0), which means that the manuscript, images, and Supporting Information files will be freely available online, and any third party is permitted to access, download, copy, distribute, and use these materials in any way, even commercially, with proper attribution. For these reasons, we cannot publish previously copyrighted maps or satellite images created using proprietary data, such as Google software (Google Maps, Street View, and Earth). For more information, see our copyright guidelines: http://journals.plos.org/plosone/s/licenses-and-copyright.

Reviewers' comments:

Reviewer's Responses to Questions

**Comments to the Author**

1. Is the manuscript technically sound, and do the data support the conclusions?

Reviewer #1: Yes

Reviewer #2: Yes

2. Has the statistical analysis been performed appropriately and rigorously? 

Reviewer #1: Yes

Reviewer #2: Yes

3. Have the authors made all data underlying the findings in their manuscript fully available?

Reviewer #1: Yes

Reviewer #2: Yes

4. Is the manuscript presented in an intelligible fashion and written in standard English?

Reviewer #1: Yes

Reviewer #2: Yes

5. Review Comments to the Author

Reviewer #1: This paper focuses on analyzing the VOCs that have been found in Hawaií and performs an analysis of the VOC variants to determine their point of origin. The authors also analyze the case numbers during quarantine and post quarantine in an attempt to demonstrate the efficacy of quarantine in delaying the entry of VOC to Hawaií, which was, (unsurprisingly) confirmed through analysis and comparison with Utah.

The strength of the paper is the thoroughness of the analysis of the genomic data available for the VOCs in Hawaii and the comparison to the VOCs worldwide from banked genomic data. The methods and the analysis of the genomic data is very well presented and explained.

What I feel the authors could improve is the background information to help the general reader better understand the significance and impact of the data presented. In particular, I would suggest that the authors rewrite the introduction to describe the general epidemiological trends of COVID infection in Hawaií, and also provide more details of what is meant by ‘quarantine’ as this has different guidelines in different countries. This would then provide the readers with a better background heading into the core findings and be able to better appreciate the findings. In the intro line 52 to 63 reads like content better suited to the discussion than the introduction?

I would also suggest that in terms of the discussion there are a few other points the authors may wish to briefly mention in the writing and discussion – it is of course of great epidemiological significance to identify the source of infection to understand the pattern of infection and global spread, however I would argue that the authors assertions (line 467) that the source of infection must be ascertained before steps can be taken may be overstating the case as by that time the case is already in, and it may be more appropriate to argue that understanding the origin of cases (eg highest from California) may be a reason to look at the processes in that country or in the infection control measures in place in that country for review? With regard to line 367 where the authors have highlighted that the highest number came from California I would suggest inferring some suggestions as to why – did California have different regulations on COVID control? Or was it because more people entering Hawaií were from California? Ideas about this then provide more guidance to public health measures at appropriate points in the chain of transmission. In addition, while limiting case numbers is of paramount concern, economic and social considerations also are a factor in deciding on the measure to implement - meaning the data and statistics here are a key consideration, but they are not the only ones.

Reviewer #2: dear author,

this paper is much appreciable and it gives the origin and spread of variants of SARS Cov-2 in different areas.

methods should have been simplified with flowchart or something. then it would be easy to reproduce by some other .

6. PLOS authors have the option to publish the peer review history of their article (what does this mean?). If published, this will include your full peer review and any attached files.

Reviewer #1: **Yes: **Priyia Pusparajah

Reviewer #2: No

---

## [Author Response · Author response to Decision Letter 0]

18 Oct 2022

Editor Comments 1 (09/26/2022):

Comment 1:

1. Thank you for your response regarding the potential copyright of your Figures. We note that you have contacted the copyright holder directly. Given what we have seen in the postscript, their approval should be enough to proceed.

At this time, please upload screenshots of your email correspondence with the copyright holder with your resubmission, and this should be good to proceed.

Response 1:

We thank the editors for this comment. Figures generated with the usmap package have been restored and the email correspondence with the copyright holder has been included.

Editor Comments 2 (09/09/2022): 

Comment 1:

1. We note that the grant information you provided in the ‘Funding Information’ and ‘Financial Disclosure’ sections do not match.

When you resubmit, please ensure that you provide the updated Funding Information.

Response 1: 

We thank the editor for this comment. The Funding Information has been updated to match the Financial Disclosure.

Comment 2:

2. We note that several of your files are duplicated on your submission. Please remove any unnecessary or old files from your revision, and make sure that only those relevant to the current version of the manuscript are included.

Response 2: 

We thank the editor for this comment. The duplicate files have been removed. The old files have been removed. 

Comment 3:

3. Thank you for your response regarding the potential copyright of your Figures. Unfortunately, at this time, it appears that the package usmap uses the GPL license, which is not compatible with our CC-BY 4.0 license. As such, please note the below prompts:

A. Was the usmap package used for both Figure 2 and Figure 3?

B. For any Figure that used the usmap package, we will require specific consent from the copyright holder to publish these images in PLOS ONE, under the CC BY 4.0 license. To seek permission from the copyright owner to publish your map figures under the specific Creative Commons Attribution License (CCAL), CC BY 4.0, please contact them with the following text and PLOS ONE Request for Permission form (http://journals.plos.org/plosone/s/file?id=7c09/content-permission-form.pdf):

“I request permission for the open-access journal PLOS ONE to publish XXX under the Creative Commons Attribution License (CCAL) CC BY 4.0 (http://creativecommons.org/licenses/by/4.0/). Please be aware that this license allows unrestricted use and distribution, even commercially, by third parties. Please reply and provide explicit written permission to publish XXX under a CC BY license.”

Please upload the granted permission to the manuscript as an other file. In the figure caption of the copyrighted figure, please include the following text: “Republished from [ref] under a CC BY license, with permission from [name of publisher], original copyright [original copyright year].”

Please note that RightsLink permission forms often impose use restrictions that are incompatible with our CC BY 4.0 license, and we are therefore unable to accept these permissions. For this reason, we strongly recommend contacting copyright holders with the PLOS ONE Request for Permission form.

If you are unable to obtain permission from the copyright holder, please either A) remove the figure or B) supply a replacement figure that complies with the CC BY 4.0 license. Please check copyright information on all replacement figures and update the figure caption with source information.

USGS National Map Viewer (http://viewer.nationalmap.gov/viewer/)

USGS Earth Resources Observatory and Science (EROS) Center (http://eros.usgs.gov/#)

The Gateway to Astronaut Photography of Earth (https://eol.jsc.nasa.gov/)

Maps at the CIA (https://www.cia.gov/library/publications/the-world-factbook/docs/refmaps.html)

NASA Earth Observatory (http://earthobservatory.nasa.gov/)

Landsat (http://landsat.visibleearth.nasa.gov/)

Natural Earth (http://www.naturalearthdata.com/)

Response 3:

We thank the editor for this comment. A) Yes, the usmap package was used for both figure 2 and 3. We have removed all images generated with the usmap package and have replaced the images with those acquired from the recommended Natural Earth (http://www.naturalearthdata.com/). 

Editor Comments 3 (07/15/2022):

Comment 1:

Response 1:

We thank the editor for this recommendation. We have submitted the protocol to protocols.io with the following DOI: dx.doi.org/10.17504/protocols.io.x54v9yqz4g3e/v1 (Private link for reviewers: https://www.protocols.io/private/3136D4A315E611ED832E0A58A9FEAC02 to be removed before publication.). We will release the protocol publicly after the manuscript is accepted for publication.

Comment 2:

Response 2:

We thank the editor for this comment. The manuscript has been updated to meet PLOS ONE style requirements. 

Comment 3:

2. We note that Figure 2 in your submission contain map images which may be copyrighted. All PLOS content is published under the Creative Commons Attribution License (CC BY 4.0), which means that the manuscript, images, and Supporting Information files will be freely available online, and any third party is permitted to access, download, copy, distribute, and use these materials in any way, even commercially, with proper attribution. For these reasons, we cannot publish previously copyrighted maps or satellite images created using proprietary data, such as Google software (Google Maps, Street View, and Earth). For more information, see our copyright guidelines: http://journals.plos.org/plosone/s/licenses-and-copyright.

Response 3:

We thank the editor for this comment and recommendations. We have opted to replace the image and have produced the images ourselves using open-source R with usmap and ggplot2 packages.

Comment 4:

Response 4:

We thank the editor for this comment. All references have been either confirmed or updated.

6. O’Toole Á, Scher E, Underwood A, Jackson B, Hill V, McCrone J, et al. pangolin: lineage assignment in an emerging pandemic as an epidemiological tool. In: PANGO lineages [Internet]. 2021 [cited 11 Mar 2021]. Available: github.com/cov-lineages/pangolin

Has been published since our original submission and has been replaced with:

6. O’Toole Á, Scher E, Underwood A, Jackson B, Hill V, McCrone JT, et al. Assignment of epidemiological lineages in an emerging pandemic using the pangolin tool. Virus Evolution. 2021;7: veab064. doi:10.1093/ve/veab064

16. R Core Team. R: A language and environment for statistical ## computing. [Internet]. Vienna, Austria: R Foundation for Statistical Computing; 2020. Available from: https://www.R-project.org/

Has been updated to: 

16. R Core Team. R: A language and environment for statistical computing. Vienna, Austria: R Foundation for Statistical Computing; 2020. Available: https://www.R-project.org/

17. Wickham H. ggplot2: Elegant Graphics for Data Analysis. 2nd ed. 2016. Cham: Springer International Publishing : Imprint: Springer; 2016. 1 p. (Use R!).

Has been updated to:

17. Wickham H. ggplot2: Elegant Graphics for Data Analysis. Springer-Verlag New York; 2016. Available from: https://ggplot2.tidyverse.org

28. Scobie H. Update on Emerging SARS-CoV-2 Variants and Vaccine Considerations. 2021 May 12;30.

Has been updated to:

28. Scobie H. Update on Emerging SARS-CoV-2 Variants and Vaccine Considerations. 2021 May 12. Available from: https://www.cdc.gov/vaccines/acip/meetings/downloads/slides-2021-05-12/10-COVID-Scobie-508.pdf

33. Jangra S, Ye C, Rathnasinghe R, Stadlbauer D, PVI study group, Krammer F, et al. The E484K mutation in the SARS-CoV-2 spike protein reduces but does not abolish neutralizing activity of human convalescent and post-vaccination sera. Infectious Diseases (except HIV/AIDS); 2021 Jan. doi:10.1101/2021.01.26.21250543

Has been published since our original submission and has been replaced with:

33. Jangra S, Ye C, Rathnasinghe R, Stadlbauer D, Personalized Virology Initiative study group, Krammer F, et al. SARS-CoV-2 spike E484K mutation reduces antibody neutralisation. Lancet Microbe. 2021;2: e283–e284. doi:10.1016/S2666-5247(21)00068-9

35. Deng X, Garcia-Knight MA, Khalid MM, Servellita V, Wang C, Morris MK, et al. Transmission, infectivity, and antibody neutralization of an emerging SARS-CoV-2 variant in California carrying a L452R spike protein mutation. medRxiv. 2021; 2021.03.07.21252647. doi:10.1101/2021.03.07.21252647

Has been published since our original submission and has been replaced with:

35. Deng X, Garcia-Knight MA, Khalid MM, Servellita V, Wang C, Morris MK, et al. Transmission, infectivity, and neutralization of a spike L452R SARS-CoV-2 variant. Cell. 2021;184: 3426-3437.e8. doi:10.1016/j.cell.2021.04.025

36. Li Q, Wu J, Nie J, Zhang L, Hao H, Liu S, et al. The Impact of Mutations in SARS-CoV-2 Spike on Viral Infectivity and Antigenicity. Cell (Cambridge). 2020;182: 1284-1294.e9. doi:10.1016/j.cell.2020.07.012

Has been updated to:

36. Li Q, Wu J, Nie J, Zhang L, Hao H, Liu S, et al. The Impact of Mutations in SARS-CoV-2 Spike on Viral Infectivity and Antigenicity. Cell. 2020;182: 1284-1294.e9. doi:10.1016/j.cell.2020.07.012

38. Aragón TJ, Newsom G. California Department of Public Health - Health Alert: Concerns re: the Use of Bamlanivimab Monotherapy in the Setting of SARS-CoV2 Variants. 2021; 4.

Has been updated to:

38. Aragón TJ, Newsom G. California Department of Public Health - Health Alert: Concerns re: the Use of Bamlanivimab Monotherapy in the Setting of SARS-CoV2 Variants. 2021. Available from: http://publichealth.lacounty.gov/eprp/lahan/alerts/CAHANBamlanivimabandSARSCoV2Variants.pdf

The following has been removed due to being revoked:

39. Moruf A. Fact Sheet For Health Care Providers Emergency Use Authorization (Eua) Of Bamlanivimab. 2021; 26.

Reviewers' comments:

Reviewer's Responses to Questions

Comments to the Author

1. Is the manuscript technically sound, and do the data support the conclusions?

Reviewer #1: Yes

Reviewer #2: Yes

2. Has the statistical analysis been performed appropriately and rigorously?

Reviewer #1: Yes

Reviewer #2: Yes

3. Have the authors made all data underlying the findings in their manuscript fully available?

Reviewer #1: Yes

Reviewer #2: Yes

4. Is the manuscript presented in an intelligible fashion and written in standard English?

Reviewer #1: Yes

Reviewer #2: Yes

5. Review Comments to the Author

Reviewer 1:

Comment 1:

Reviewer #1: This paper focuses on analyzing the VOCs that have been found in Hawaií and performs an analysis of the VOC variants to determine their point of origin. The authors also analyze the case numbers during quarantine and post quarantine in an attempt to demonstrate the efficacy of quarantine in delaying the entry of VOC to Hawaií, which was, (unsurprisingly) confirmed through analysis and comparison with Utah.

The strength of the paper is the thoroughness of the analysis of the genomic data available for the VOCs in Hawaii and the comparison to the VOCs worldwide from banked genomic data. The methods and the analysis of the genomic data is very well presented and explained.

Response 1:

We thank the reviewer for these comments.

Comment 2:

What I feel the authors could improve is the background information to help the general reader better understand the significance and impact of the data presented. In particular, I would suggest that the authors rewrite the introduction to describe the general epidemiological trends of COVID infection in Hawaií, and also provide more details of what is meant by ‘quarantine’ as this has different guidelines in different countries. This would then provide the readers with a better background heading into the core findings and be able to better appreciate the findings. In the intro line 52 to 63 reads like content better suited to the discussion than the introduction?

Response 2:

We thank the reviewer for these comments. In the revised manuscript, lines 52 to 63 have been moved to the discussion and we have addressed the remaining comments as follows:

“Hawaii has experienced unique epidemics within the coronavirus disease 2019 (COVID-19) pandemic, in that Pacific Islanders, which account for 4% of the population, once accounted for nearly 30% of COVID-19 cases.(1) Further, the Japanese population of Hawaii currently accounts for 6% of the population and experiences 15% of COVID-19 cases. White persons, in contrast, account for 37% of the population and 25% of the cases.(2) As such, a heightened need exists to understand SARS-CoV-2 introduction into Hawaii and the effect of public policy measures. Early in the pandemic, in an attempt to control the spread of severe acute respiratory syndrome coronavirus 2 (SARS-CoV-2), Hawaii, like 42 other states in the United States, implemented a quarantine defined by “Stay-at-Home” orders. State-at-Home orders directed residents to stay inside homes except for essential needs and closed operations of non-essential businesses.(3) In addition to this public policy, more than 22,300 SARS-CoV-2 sequences submitted to GISAID and GenBank originate from Hawaii to facilitate further studies.”

Comment 3:

I would also suggest that in terms of the discussion there are a few other points the authors may wish to briefly mention in the writing and discussion – it is of course of great epidemiological significance to identify the source of infection to understand the pattern of infection and global spread, however I would argue that the authors assertions (line 467) that the source of infection must be ascertained before steps can be taken may be overstating the case as by that time the case is already in, and it may be more appropriate to argue that understanding the origin of cases (eg highest from California) may be a reason to look at the processes in that country or in the infection control measures in place in that country for review? 

Response 3:

We thank the reviewer for this discussion and argument. We have made the statement less assertive and included the reasoning provided in this comment in the revised manuscript as follows: 

“Policy-makers should first ascertain the source of the spread before they can control and limit the spread of future VOC. By understanding the source responsible for the highest number of cases, policy-makers can look at interactions between that area and the host area, the policies in that area, identify the reasons for the spread, and address those reasons with appropriate measures both in the present and in future COVID-19 waves.”

Comment 4:

With regard to line 367 where the authors have highlighted that the highest number came from California I would suggest inferring some suggestions as to why – did California have different regulations on COVID control? Or was it because more people entering Hawaií were from California? Ideas about this then provide more guidance to public health measures at appropriate points in the chain of transmission. In addition, while limiting case numbers is of paramount concern, economic and social considerations also are a factor in deciding on the measure to implement - meaning the data and statistics here are a key consideration, but they are not the only ones.

Response 4:

We thank the reviewer for this comment and suggestion. We have addressed this in the revised manuscript as follows:

In 2020, 27% of all travelers to Hawai’i originated from California, with 53% coming from the West Coast. Further, Hawai’i residents traveled to the West Coast, specifically Las Vegas, Nevada.(1,28,29) 

However, the following is additional information:

“From the analysis of the SARS-CoV-2 sequence data, a policy-maker could reasonably consider focusing on additional screening, contact tracing, and quarantine efforts among visitors and residents arriving from and traveling to the West Coast of the continental United States. There are several possible reasons for this vast majority of SARS-COV-2 influx from the US West Coast. In 2020, 27% of all travelers to Hawaii originated from California, with 53% coming from the West Coast. California's biggest domestic traveling demographic is in-state travel, meaning that the state likely spreads SARS-CoV-2 efficiently and uniformly within California.(44) Second, Hawaii residents traveling to the West Coast and returning home once infected with the virus. The first case of COVID-19 in Hawaii and the first case of the Delta variant were brought to Hawaii by residents (both vaccinated and unvaccinated) returning from travel (from Mexico and Nevada, respectively).(45–47) Additionally, 62% of early cases in Hawaii were in either visitors to Hawaii or returning residents.(47) There are presumably additional factors that participated in the 76% of SARS-CoV-2 VOC attributable to California. Regardless, policymakers must evaluate these possible collective factors and social and economic implications together to determine the appropriate public-policy action.” 

Reviewer 2:

Reviewer #2: dear author,

Comment 1:

this paper is much appreciable and it gives the origin and spread of variants of SARS Cov-2 in different areas.

Response 1:

We thank the reviewer for this comment. 

Comment 2:

methods should have been simplified with flowchart or something. then it would be easy to reproduce by some other .

Response 2:

We thank the reviewer for this comment and have added a flowchart as a figure and have uploaded the method to protocols.io. 

---

## [Editor Report · Decision Letter 1]

15 Nov 2022

Genomic Analysis of SARS-CoV-2 Variants of Concern Circulating in Hawai’i to Facilitate Public-Health Policies

PONE-D-21-19856R1

Dear Dr. Maison,

We’re pleased to inform you that your manuscript has been judged scientifically suitable for publication and will be formally accepted for publication once it meets all outstanding technical requirements.

Kind regards,

Ming Zhang

Academic Editor

PLOS ONE
---

## [Editor Report · Acceptance letter]

21 Nov 2022

PONE-D-21-19856R1 

Genomic Analysis of SARS-CoV-2 Variants of Concern Circulating in Hawai’i to Facilitate Public-Health Policies 

Dear Dr. Maison:

I'm pleased to inform you that your manuscript has been deemed suitable for publication in PLOS ONE. Congratulations! Your manuscript is now with our production department. 

Kind regards, 

on behalf of

Dr. Ming Zhang 

Academic Editor

PLOS ONE